# High-Throughput Quantitative Screening of Glucose-Stimulated Insulin Secretion and Insulin Content Using Automated MALDI-TOF Mass Spectrometry

**DOI:** 10.3390/cells12060849

**Published:** 2023-03-09

**Authors:** Clément Philippe Delannoy, Egon Heuson, Adrien Herledan, Frederik Oger, Bryan Thiroux, Mickaël Chevalier, Xavier Gromada, Laure Rolland, Philippe Froguel, Benoit Deprez, Sébastien Paul, Jean-Sébastien Annicotte

**Affiliations:** 1Univ. Lille, Inserm, CHU Lille, Institut Pasteur de Lille, CNRS, U1283–UMR 8199–EGID, F-59000 Lille, France; 2Univ. Lille, CNRS, Centrale Lille, Univ. Artois, UMR 8181–UCCS—Unité de Catalyse et de Chimie du Solide, F-59000 Lille, France; 3Univ. Lille, Inserm, Institut Pasteur de Lille, U1177—Drugs and Molecule for Living Systems, F-59000 Lille, France; 4Univ. Lille, UMRt BioEcoAgro 1158-INRAE, Équipe Métabolites Secondaires D’origine Microbienne, Institut Charles Viollette, F-59000 Lille, France; 5Univ. Lille, Inserm, CHU Lille, Institut Pasteur de Lille, U1167-RID-AGE—Facteurs de Risque et Determinants Moléculaires des Maladies liées au Vieillissement, F-59000 Lille, France

**Keywords:** insulin secretion, pancreatic beta cell, high-throughput screening, MALDI-TOF mass spectrometry, type 2 diabetes

## Abstract

Type 2 diabetes (T2D) is a metabolic disorder characterized by loss of pancreatic β-cell function, decreased insulin secretion and increased insulin resistance, that affects more than 537 million people worldwide. Although several treatments are proposed to patients suffering from T2D, long-term control of glycemia remains a challenge. Therefore, identifying new potential drugs and targets that positively affect β-cell function and insulin secretion remains crucial. Here, we developed an automated approach to allow the identification of new compounds or genes potentially involved in β-cell function in a 384-well plate format, using the murine β-cell model Min6. By using MALDI-TOF mass spectrometry, we implemented a high-throughput screening (HTS) strategy based on the automation of a cellular assay allowing the detection of insulin secretion in response to glucose, i.e., the quantitative detection of insulin, in a miniaturized system. As a proof of concept, we screened siRNA targeting well-know β-cell genes and 1600 chemical compounds and identified several molecules as potential regulators of insulin secretion and/or synthesis, demonstrating that our approach allows HTS of insulin secretion in vitro.

## 1. Introduction

Type 2 Diabetes (T2D) is characterized by high blood glucose levels and develops due to inadequate pancreatic β-cell function (i.e.*,* insulin secretion) and peripheral insulin resistance. In Europe, the global prevalence of diabetes is currently estimated at 8% of the population, with T2D representing about 90% of cases [1]. Although lifestyle modification is the first reference treatment for T2D, modifying the patient’s habits is often ineffective to stabilize glycemia and combinations of pharmacological treatments (metformin, sulfonylurea, incretin enhancers, GLP-1 mimetics, etc.) are often required to treat T2D [2,3]. Yet, long-term control of glycemia remains a challenge for most patients, particularly when β-cell capacity to secrete insulin decreases with age [4].

In the context of research aimed at discovering new therapeutic targets for T2D, the identification of new natural or synthetic compounds or target genes that have the capacity to restore or increase insulin secretion and content is essential. Therefore, developing high-throughput screening (HTS) strategies to rapidly measure insulin secretion or synthesis is crucial to address the unmet need of new efficient molecules for T2D patients. Currently, several strategies have been developed based on assays aiming at measuring secreted hormones or specific proteins (e.g., insulin or c-peptide) [5], mainly through Enzyme-Linked Immuno Sorbent Assay (ELISA) approaches. However, this technique, although widely used in academic research and clinical diagnosis, has several limitations, such as the technical detection of the target protein, its precise quantification, the duration of the experiment and the costs of the reagents. Altogether, these limitations preclude the use of ELISA strategies for real HTS approaches to evaluate the action of a large collection of potential drugs that can stimulate insulin secretion.

Along with ELISA, complementary techniques for the high-throughput screening of insulin secretion have recently emerged with a high potential for T2D therapy. To overcome the experimental and technical limits of ELISA, several laboratories have developed genetically engineered cellular tools to measure β-cell function. In these modified cells, the use of immunofluorescence microscopy or luciferase-based approaches allows the measurement of a modified C-peptide [5,6,7,8,9]. Although easy to handle, these genetically modified tools partially address the functionality of the β-cell since they do not directly assess the level of production of endogenous insulin content or the secretion of insulin in response to a physiological stimulation such as glucose. Therefore, to tackle these limitations, we developed a new strategy aiming at implementing an automated approach that could allow HTS of drugs that efficiently modulate glucose-stimulated insulin secretion. This automated process, using mass spectrometry to directly measure insulin, is not only faster but also less expensive, enabling for the first time the implementation of high-throughput exploratory strategies to identify new biological targets or bioactive compounds through the screening of chemical or siRNA libraries. Here, we describe a high-throughput screening approach based on matrix-assisted laser desorption/ionization time-of-flight mass spectrometry (MALDI-TOF) to quantify insulin production and secretion in a mouse model of β-cells, the Min6 cell line. We automated, in a 384-well format, siRNA-based loss-of-function of candidate genes to study their effect on insulin secretion. Finally, as a proof of concept, we screened more than 1600 chemical compounds and identified several drugs that modulate insulin secretion or content.

## 2. Materials and Methods

### 2.1. Cell Culture and Treatments

MIN6 cells (Addexbio) were maintained in 25 mM glucose, Glutamax DMEM medium (Gibco, 31966-021, Waltham, MA, USA), supplemented with 15% heat-inactivated fetal bovine, 100 μg/mL penicillin-streptomycin and 55 μM beta-mercaptoethanol (Gibco) and cultured in a humidified atmosphere with 5% CO_2_ at 37 °C. Cells were seeded at 20,000 cells/well using a Multidrop Combi dispenser (Thermo Fisher Scientific, Waltham, MA, USA) into 384-well plate black µclear. Cells were then treated with different chemical compounds and/or siRNA, as described below (see Appendix A for the list of siRNA used in this study).

### 2.2. Automated Glucose-Stimulated Insulin Secretion (GSIS) in 384-Well Format

For GSIS experiments, Min6 cells were plated in a 384-well plate (2.104 cells/well) and incubated in 80 µL of starvation buffer Krebs Ringer buffer (KRB) [NaCl 115 mM; KCl 4.7 mM; CaCl_2_ 2.6 mM; NaHCO_3_ 20 mM; MgSO_4_ 1.2 mM; KH_2_PO_4_ 1.2 mM; HEPES 16 mM] supplemented with 0.5% fatty-acid free BSA for 1 h at 37 °C and 5% CO_2_. The automated GSIS pipeline protocol started on the BioCel platform system (Agilent, Santa Clara, CA, USA) including integrated devices for incubation, washing, distribution, pipetting the liquid handling system, microplate stacking or sealing systems. All the tasks were sequentially operated on each device using the Direct Drive Robot (DDR) arm (Agilent Biotechnologies) positioned at the center of the platform. At the beginning of the run, microplates placed at 37 °C in 5% CO_2_ atmosphere in the Incubator (Liconic, Mauren, Liechtenstein) were conveyed via a telescopic lift arm on the deck of the platform. The cells were first washed from their culture medium by 4 washes of KRB-BSA with the EL406 washer distributor liquid handling device (Biotek, Agilent, Santa Clara, CA, USA) suitable for 384-well plates. Control of the dispense (with tubes angled to 20°) and aspiration of liquid due to the dual-action manifold system of the device permitted the complete reduction of the loose of cell layers during this washing step. The cells were then incubated for 1 h at 37 °C in KRB-BSA for the starvation stage. To avoid BSA interferences in MALDI-TOF mass spectrometry analysis, Min6 cells were washed 5 times with BSA-free KRB buffer supplemented with 2.8 mM glucose after starvation using a Biotek washing robot. After 1 h at 37 °C, cells were washed 5 times with KRB buffer and supplemented with 2.8 mM glucose by distribution of a solution of KRB-glucose using the disposable peristaltic pump cassette system (EL406 washer distributor), for 1 h at 37 °C and 5% CO_2_. Then, 40 µL of the supernatant was collected for insulin quantification in 2.8 mM glucose conditions using a Bravo liquid handler and cells were subsequently incubated in 80 µL of KRB buffer containing 20 mM glucose for 1 h at 37 °C and 5% CO_2_. A 100 µL sample of the supernatant was collected for insulin quantification in 20 mM glucose conditions. The intracellular insulin content was recovered in 40 µL of lysis buffer containing 75% ethanol and 1.5% HCl. Microplates with all collected samples were sealed with a PlateLoc Thermal Microplate Sealer (Agilent Technologies) after the collecting step with Bravo and frozen at −80 °C until analysis. Insulin concentration was measured through ELISA according to the manufacturer’s instructions (Mercodia, Uppsala, Sweden) or mass spectrometry as described below. Samples were frozen at −80 °C before further processing.

### 2.3. Automated siRNA Reverse Transfection in 384-Well Format

To validate a potential functional effect on GSIS, we selected two siRNA targeting genes previously known to control insulin secretion (ON-TARGETplus SMARTpool siRNA targeting mouse Pdx1 (L-040402-01-0005) and mouse Kcnj11 (L-042183-00-0005)) and two controls (non-targeting negative controls and siGLO fluorescent control). siRNA transfection was performed by reverse transfection using 0.375% Dharmafect1 transfection reagent (GE Dharmacon) and 50 nM siRNA. siRNAs (200 nL of a 20 µM stock; 0.16 pmol/well) were dispensed into 384-well assay plates (Greiner Bio-One; 78109) using an Echo 550 Series Liquid Handler (Labcyte, San José, CA, USA). On the day of assay, plates that contained siRNAs were thawed and equilibrated to room temperature. Dharmafect1 transfection reagent (0.3 µL/well in DMEM media, Gibco) was added to the assay plates using a Multidrop Combi dispenser (Thermo Fisher Scientific, Waltham, MA, USA), with a standard tube dispensing cassette. After 20 min of incubation at room temperature, cells (65 µL of 300,000 cells/mL; 20,000 cells/well) were added to the assay plates with the Multidrop Combi dispenser and standard tube dispensing cassette. Assay plates were incubated at 37 °C, 5% CO_2_ in a controlled-humidity incubator and cultured for 48 h before GSIS assays. Samples were frozen at −80 °C before further processing.

### 2.4. Automated Incubation with Chemical Compounds

Min6 cells (80 µL of 250,000 cells/mL; 20,000 cells/well) were added to the assay plates with the Multidrop Combi dispenser (ThermoFisher) and standard tube dispensing cassette. Assay plates were incubated at 37 °C, 5% CO_2_ in a controlled-humidity incubator during 24 h. Repaglinide (Euromedex—R1426, Souffelweyersheim, France) and diazoxyle (CliniSciences—BG0437, Nanterre, France) were used at a concentration of 100 nM and 100 µM. A library containing 1600 lead-like molecules was obtained from Asinex company (Moscow, Russia). The compounds were dissolved at a concentration of 10 mM in DMSO and one batch of this library was distributed in a 384-well LDV microplate (LP-0200, Beckman coulter, Pasadena, CA, USA). For the automated screening assay, 200 nL of 1600 Asinex compounds (5 microplates) or reference products were transferred in intermediate microplates (Greiner Bio-One; 78109, Vilvoorde, Belgium) from a source 384-well LDV microplate using a nanoacoustic transfer device (Echo 550), then stored at −20 °C until use. Eighteen hours before the automated GSIS protocol, using the BioCel platform, intermediate microplates were filled with 60 µL of cell culture medium with an EL406 washer distributor (Cassette for distribution). Then, cell microplates previously placed in the incubator were sent to the EL406 washer distributor for the aspiration step in order to leave a remaining volume of 40 µL per well. On the deck of the Bravo device, compounds diluted in medium in the intermediate microplate were mixed 4 times and a volume of 20 µL was transferred in the cell microplate for a treatment at the final concentration of 10 µM. Cells were treated overnight with different compounds and samples were frozen at −80 °C before further processing.

### 2.5. Automated MALDI-TOF Mass Spectrometry Analysis

The MALDI target were prepared in an automated way using a Biomek NX^p^ liquid handler (Beckman Coulter, Fullerton, CA, USA). The detailed robot routine is available in the Appendix A. First, the 384-well microplate (Greiner) containing the previously prepared samples was placed onto the robot deck, along with the other required labware. Then, for cellular content samples only, 40 µL of MS-grade water was added to the 40 µL of samples to dilute it by a factor of 2. The dilution was performed by mixing 10 µL of the resulting solution 3 times. Then, 30 µL of a stock solution of bovine insulin (Sigma-Aldrich, Saint Louis, MO, USA) at 10 μg.mL^−1^ in 3 mM hydrochloric acid (pH 2.5) (for content samples, 5 μg.mL^−1^ for extracellular high and low glucose samples) was separated into a second 384-well microplate. In parallel, 15 µL of the MALDI matrix solution containing sinapic acid (Sigma-Aldrich) at 10 mg.mL^−1^ in 50/49/1:acetonitrile/MS-grade water/trifluoroacetic acid (Sigma Aldrich) was added in each well of a third separate 384-well microplate. The different samples and solutions were then mixed in the following order: First, 10 µL of the bovine insulin solution was transferred into the microplate containing the samples and the two solutions were mixed by aspirating/refouling 10 µL of the mixture 10 times. Then, 15 µL of the resulting mixture was transferred into the plate containing the matrix and the two solutions were mixed by aspirating/refouling 10 µL of the mixture 10 times. Following this, 2 µL of the resulting sample/bovine insulin/matrix mixture was immediately deposited as a drop on each spot of a MALDI MTP384 Polish steel target (Bruker Daltonics, Bremen, Germany). This was made possible by the creation and 3D printing of an adapter to allow the use of the MALDI target by the robot. Its design is presented in the Appendix A. The drops were then dried at room temperature, and the MALDI target was introduced in the source chamber of an Autoflex Speed (Bruker Daltonics, Bremen, Germany). All MALDI-TOF analyses were performed in linear positive mode using an in-house method for insulin detection (LP_5-20_kDa.par), created from the manufacturer’s automatic method LP_5-20_kDa.par. Equipment parameters were as follows: voltage values of ion sources #1 and #2 set as 19.00 and 16.50 keV, respectively; voltage values of reflectron #1 and #2 set as 21.00 and 9.50 keV, respectively; lens tension 8.00 keV; pulsed extraction 120 ns; laser intensity between 60 and 70%; laser global attenuator offset set to 41%; attenuator offset set to 32%; attenuator range set to 25%; detector gain voltage set to 2600 V (+300 V boost); Smartbeam parameter set to ultra and sample rate and digitizer settings set to 4.00 GS/s. The MS signals were acquired by summing 5000 laser shots per spectrum. Prior to each analysis, the spectrometer was calibrated using the monoisotopic values of the manufacturer’s Protein Calibration Standard I calibrant (Bruker Daltonics, Billerica, MA, USA), containing insulin ([M+H]^+^—*m*/*z* = 5734.51), ubiquitin I ([M+H]^+^—*m*/*z* = 8565.76), cytochrome C ([M+H]^+^—*m*/*z* = 12,360.97), myoglobin ([M+H]^+^—*m*/*z* = 16,952.30), cytochrome C ([M+2H]^2+^—*m*/*z* = 6180.99) and myoglobin ([M+2H]^2+^—*m*/*z* = 8476.65). The calibrant was prepared by mixing 5 µL of calibrant diluted mixture prepared according to manufacturer’s specification with 5 µL of a 10 mg.mL^−1^ HCCA matrix in 50/49.9/0.1:acetonitrile/MS-grade water/trifluoroacetic acid, and 2 µL were then spotted on a Polished Steel 384 MALDI MTP target. Mass spectra of the sample were first visualized using FlexAnalysis software (version 3.4; Bruker Daltonics, Billerica, MA, USA), after their calibration. To produce a comprehensive document, providing the intensity and area ratio of the murine and bovine insulin, as well quality control, in a way that is simple and easily understood by non-specialist experimenters, a short program was coded. It was written in VBA (code available in the Appendix A), and linked to an Excel sheet (Microsoft, Redmond, WA, USA). After an automated export for the peak list from the mass spectra using FlexAnalysis software into a dedicated Excel sheet, this program can first perform an automated quality control, based on the intensity and the mass of the bovine insulin. Then, it automatically detects the murine insulin and performs the ratio in area or intensity compared to bovine insulin, depending on the user’s preference. Finally, it sorts up all information and presents it in a clear colored view as a plate map, associated with ratio values and corresponding bar charts to quickly see the compounds that produce the highest signal compared to the blank and others.

### 2.6. Measurement of Transfection Efficiency by Flow Cytometry

To assess the transfection efficiency, Min6 cells were transfected with a siGLO (Green Transfection Indicator D-001630-01-05, Dharmacon, Lafayette, CO, USA) labelled with fluorescein (CF/FAM/FITC)), using different concentrations of siGLO and transfection reagent. Transfection efficiency was determined 48 h after transfection using flow cytometry. Cells were acquired on a BD LSR Fortessa flow cytometer, and fluorescence of transfected Min6 cells was quantified by flow cytometry.

### 2.7. Immunofluorescence

Images from transfected Min6 cells with siGLO and labeled with Hoechst 33342 (40 ng/mL, Thermofisher) were acquired with the microscope IN Cell Analyser 6000 (GE Healthcare Life Sciences, Buc, France) in 10× magnification in non-confocal mode with a DAPI filter set (ex.405/em.455 nm) and an FITC filter set (ex.488/em.525 nm) on the automated high content screening platform (Agilent Technologies, Santa Clara, CA, USA) (Equipex Imaginex Biomed, Institut Pasteur de Lille, France).

### 2.8. RNA Extraction and Quantitative Real-Time Polymerase Chain Reaction (qRT-PCR)

Total RNA was extracted from Min6 cells using the RNeasy Plus Microkit (Qiagen, Hilden, Germany) following manufacturer’s instructions. Gene expression was measured after reverse transcription by quantitative real-time PCR (qRT-PCR) with FastStart SYBR Green master mix (Roche) using a LightCycler Nano or LC480 instruments (Roche, Basel, Switzerland). qRT-PCR results were normalized to endogenous cyclophilin reference mRNA levels as previously described [10]. The results are expressed as the relative mRNA level of a specific gene expression using the formula 2^−ΔCt^. The complete list of primers is presented in Appendix A.

### 2.9. Immunoblotting Experiments

Cells were washed with cold PBS and lysed in radioimmunoprotein-assay (RipA) buffer (10 mM Tris/HCl, 150 mM NaCl, 1% (*v*/*v*) Triton X-100, 0.5% (*w*/*v*) sodium deoxycholate, 0.1% (*w*/*v*) sodium dodecyl sulfate and protease inhibitors, pH 7.4), and maintained under constant agitation for 30 min. Cell extracts were then centrifuged at 16,000× *g* for 20 min at 4 °C). Protein concentration was determined by the BCA protein assay kit according to the manufacturer’s instructions. Equal amounts of protein were resolved on 10% SDS-PAGE under reducing conditions and proteins were transferred to a nitrocellulose membrane. Blots were incubated with primary antibodies directed against PDX1 (1:1000, Abcam, ab47267, Cambridge, UK), alpha-tubulin (1:1000, Invitrogen, 32-2700, Waltham, MA, USA), washed three times with PBS-0.05% tween and followed by incubation with secondary antibodies, directed against goat anti-mouse or rabbit HRP-conjugated (1:5000, Sigma-Aldrich). The visualization of immunoreactive bands was performed using the enhanced chemiluminescence plus Western blotting detection system (GE Healthcare). Quantification of protein signal intensity was performed by volume densitometry using ImageJ 1.47t software (NIH, Bethesda, MD, USA).

### 2.10. Statistical Analysis

Data are expressed as mean ± SEM. Statistical analysis were performed using GraphPad Prism software (version 9.3, Boston, MA, USA) with two-way ANOVA with Bonferroni’s post-test for multiple comparisons as indicated in the figure legends. Differences were considered statistically significant at *p* value < 0.05 (* < 0.05; ** < 0.01; *** < 0.001; **** < 0.0001).

For the high-throughput screens, a score similar to the statistical Z-score for each test compound was calculated using the formula: Z-score = (*x* − *μ*)/σ, where *x* is the murine insulin relative intensity from a compound-treated well, *μ* is the murine insulin relative intensity from the DMSO-treated wells on the same plate and σ is the standard deviation of the murine insulin relative intensity signals of the DMSO-treated wells across all plates.

## 3. Results

### 3.1. Automation of Glucose-Stimulated Insulin Secretion (GSIS) Assay

In order to develop an automated protocol to measure insulin secretion in the mouse insulinoma Min6 cell line, we first miniaturized the GSIS protocol in 384-well plates (Appendix A) and validated that these cells do respond to glucose stimulation in these culture conditions. Min6 cells were seeded and, 48 h later, the cells were subjected to GSIS. For MALDI-TOF mass spectrometry analysis, removing BSA from the KRB buffer is a key step to limit BSA interferences and salt contaminations. Therefore, after one hour of starvation, the cells were washed five times with BSA-free KRB buffer supplemented with 2.8 mM glucose and incubated for 1 h at 37 °C. Then, half of the supernatant was collected using a Bravo liquid handler, and the cell plates were complemented with KRB buffer containing glucose at 20 mM and incubated for 1 h at 37 °C. The high glucose fractions were then collected and cells were lysed to measure insulin content.

To confirm the efficiency of the automated protocol, GSIS samples were subjected to an ELISA assay to measure insulin secretion in response to low and high glucose concentrations. Randomly selected samples from two separate 384-well plates were measured. Our data show that Min6 cells display a secretion rate which increases significantly after glucose stimulation, which represents 3 to 5% of the total insulin content (Figure 1A) or a 6- to 8-fold increase of insulin secretion when Min6 cells were stimulated from 2.8 mM to 20 mM of glucose (Figure 1B). These data demonstrate that our GSIS protocol is functional in 384-well plates and that GSIS of Min6 cells can be fully automated in 384-well plates following our protocol.

### 3.2. Automated Analysis of GSIS in 384-Well Plates through MALDI-TOF Mass Spectrometry

Having established the miniaturized GSIS protocol, we next wanted to quantitatively detect insulin from GSIS assays through an automated approach that should be at least as sensitive, reliable and quantitative as ELISA and that can be automated. To this end, we selected MALDI-TOF mass spectrometry. First, we set up a spiked-based strategy that allows a precise quantification and normalization of the insulin protein present in our samples. Since the murine insulin has a molecular weight of 5803 Da, we selected bovine insulin as an internal standard, which has a molecular weight of 5733 Da and is assumed to present an ionizability very close to the one of murine insulin as its amino acid composition is very similar. To quantitatively measure insulin from GSIS supernatants, samples were spiked with known concentrations of bovine insulin. Following MALDI-TOF mass spectrometry, a final mass spectrum was obtained, corresponding to the sum of 5000 laser shots (Figure 2A). The relative intensity emitted by the detected murine insulin was normalized to the relative intensity of our spiked bovine insulin internal standard. Importantly, we observed that mixing our internal standard with our samples did not affect the signal strength of the detected murine insulin (Figure 2B). Then, to demonstrate the reliability of MALDI-TOF mass spectrometry to detect insulin, samples from GSIS experiments performed in 384-well plates were analyzed through an ELISA assay and data were compared to MALDI-TOF mass spectrometry results. When we measured automated GSIS through ELISA or MALDI-TOF mass spectrometry, we could not observe differences in GSIS results between these two approaches. The analysis of two independent plates demonstrated that Min6 cells secreted between six to eight times more insulin at 20 mM glucose compared to 2.8 mM glucose, independent of the method of detection, i.e., mass spectrometry or ELISA (Figure 2C). These results demonstrate a strong reliability between both detection methods to measure insulin protein and suggest that MALDI-TOF mass spectrometry is as sensitive, as reliable and as quantitative as ELISA for GSIS measurements.

### 3.3. Automated siRNA Transfection Combined with GSIS

We next wanted to apply the automated GSIS protocol to the discovery of new potential genes involved in insulin secretion. Following our experimental strategy described above, a fully automated siRNA reverse transfection protocol was implemented with the aim to detect variations of insulin secretion upon knocking-down specific genes. We first determined the optimal transfection conditions using a 6-FAM fluorescent-labeled control siRNA. This step is crucial to ensure an efficient transfection rate to potentially obtain a significant knock-down of the expression of target genes. Using siGLO as a fluorescent control to assess transfection efficiency, we could determine the optimal concentration for transfection reagents, siRNA and time of incubation (Appendix A). Once the automated miniaturized transfection and GSIS protocols were validated, we tested two positive control siRNA: *Pdx1* and *Kcnj11*, two genes previously shown to negatively modulate insulin secretion. Knock-down of *Pdx1* gene in Min6 cells was validated both at the transcriptomic and protein levels (Appendix A). Upon knock-down, GSIS samples were then analyzed by MALDI spectrometry and we detected lower relative intensities for murine insulin in cells treated with siRNA targeting *Pdx1* and *Kcnj11* (Figure 3A,B). Indeed, intensity ratios under 2.8 mM and 20 mM glucose conditions showed a ten-fold increase in insulin secretion for siControl-treated Min6 cells, whereas a six-fold increase in si*Pdx1*-treated cells and a four-fold increase in si*Kcnj11*-treated cells were observed (Figure 3C). These results were further independently confirmed through an ELISA assay where siRNAs targeting *Kcnj11* and *Pdx1* induced a significant decrease in insulin secretion after a stimulation with 20 mM glucose (Figure 3D). Again, mass spectrometry analysis further confirmed the results obtained through ELISA approaches (Figure 3E). Altogether, these data demonstrate that an automated siRNA reverse transfection protocol is efficient to evaluate potential functional effects of knocking-down genes in Min6 cells and suggest that this approach provides a robust automated platform to evaluate GSIS in HTS approaches.

### 3.4. Automated Chemical Treatment Combined with GSIS

As a proof of concept, we next wanted to implement HTS strategies using our Min6 automated GSIS protocol coupled to mass spectrometry analysis. To demonstrate the feasibility of our approach, the use of pharmacological activators (repaglinide) and inhibitors (diazoxide) of insulin secretion has been undertaken to observe insulin secretion variations in Min6 cells. After treatment of the cells with these compounds, MALDI-TOF mass spectrometry analysis was employed to measure insulin secretion in response to these drugs. Our data revealed that these compounds were effective in modulating glucose-stimulated insulin secretion (Figure 4A), as demonstrated using automated MALDI TOF mass spectrometry or ELISA assays. Indeed, upon stimulation with 2.8 mM glucose conditions, Min6 cells treated with 100 nM of repaglinide had a higher basal secretion rate than untreated cells or cells incubated with 100 μM of diazoxide (Figure 4B). Upon 20 mM glucose condition treatment, a two-fold increase and a two-fold decrease in insulin secretion in the presence of repaglinide and diazoxide, respectively, was observed (Figure 4C,D), which were again confirmed through an ELISA assays (Figure 4E). In addition, the intracellular insulin content of Min6 cells exposed to repaglinide was decreased by 30% when compared to other conditions (Appendix A), yet the total amount of insulin was similar in Min6 for all conditions of treatments (Appendix A).

As expected, cells stimulated with repaglinide showed decreased intracellular insulin content when compared to control, untreated cells. Indeed, treating Min6 cells with repaglinide induced a 70% decrease in the intracellular insulin content, and, conversely, diazoxide increased by 30% insulin content compared to control, untreated cells (Figure 5A). Thanks to our high throughput MALDI-TOF approach, we analyzed 160 vehicle-treated samples, 62 repaglinide-treated samples and 62 diazoxide-treated samples, and observed that their analysis followed a normal distribution (Figure 5B and Appendix A). In addition, we could repeat this experiment with other bioactive molecules known to induce insulin secretion, such as forskolin or IBMX, and confirmed our results obtained with repaglinide (Appendix A). After GSIS, the intensity levels corresponding to insulin observed for the cells treated with different secretagogues were markedly lower than the control, untreated cells, suggesting that these compounds may provoke intracellular insulin leakage (Appendix A).

### 3.5. Automated Library Screening to Measure Insulin Intracellular Content after GSIS

Since our automated cell culture protocol, MALDI-TOF mediated insulin detection and quantification and GSIS pipeline is adapted for high-throughput applications, we performed a pilot screening experiment in Min6 cells using a collection of 1600 compounds from ASINEX library and DMSO, repaglinide and diazoxide as controls. Here, we focused on measuring insulin content only. As observed previously, repaglinide induced a significantly lower intracellular insulin content compared to the negative controls (Figure 6).

This screening was carried out in duplicate in order to be able to calculate a z-score for all the compounds tested. The DMSO control cells presented a z-score comprised in a window between z = −1 and z = 1. The cells treated with repaglinide had a z-score between z = −1.5 and z = −3. We assumed that cells treated with compounds with the same order of magnitude of repaglinide z-score may be considered as molecules that stimulate insulin secretion. Conversely, compounds leading to a z-score close to controls or diazoxide may be considered as, at least, non-effective or inhibitors of insulin secretion. Among the 1600 compounds tested, we observed that some molecules decreased insulin content after glucose stimulation (Figure 6), suggesting that they may stimulate insulin secretion.

Amongst those molecules, we observed different compounds with a z-score close to those of repaglinide, the positive control. Altogether, our results demonstrate that our approach using MALDI-TOF mass spectrometry can be automated and efficiently used for HTS of libraries of siRNA or chemical compounds.

## 4. Discussion

Here, we describe a new automated methodology to accurately measure insulin secretion in vitro and identify potential modulators of insulin secretion in pancreatic beta cells. This new process, based on automated GSIS and MALDI-TOF mass spectrometry, was developed in a 384-well plate format to enable high-throughput analyses and identify new chemical substances or genes that potentially modulate insulin secretion or cell content. We demonstrated the efficiency of our assay by performing a pilot experiment using 1600 chemical compounds and selected siRNAs, in the mouse Min6 beta cell line. Indeed, we have developed and implemented this technological process to test the efficiency of screening different chemical compounds, but also siRNA targeting *bona fide* β-cell identity genes. Unlike other techniques such as ELISA assay, our protocol is based on direct insulin detection and not on antigen recognition. In addition, we demonstrate that the MALDI-TOF mass spectrometry strategy is as reliable as ELISA detection, while being much less expensive than existing insulin detection techniques.

High-throughput screening for insulin secretion is an approach with high potential for valorization. In the case of the insulin-secreting pancreatic β-cell, these approaches are limited by several technological constraints inherent to the methods for detection and quantification of this hormone (i.e., ELISA). To overcome these experimental limitations, several laboratories have reported the use of modified cell-based tools to indirectly measure the level of secretory activity of mouse-derived Min6 cells by fluorescence microscopy or measurement of modified peptide-C luciferase [6,7,8,9]. Although simple to manipulate, these artificial, genetically engineered tools only very partially interrogate β-cell functionality as it assesses neither endogenous insulin production nor its secretion under glucose stimulation conditions. It is with the objective of addressing these two analytical criteria that we have proposed to deploy our complete procedure of automation of the GSIS protocol followed by MALDI-TOF mass spectrometry to envision, in the future, a siRNA library (approx. 20,000 genes) and chemical library (>100,000 compounds) screening strategy in the Min6 pancreatic β-cell model. A similar study has already been undertaken on the INS-1 832/13 pancreatic beta cell model, with the screening of 1200 compounds, but unlike our study, the compounds used have already known targets [11,12].

The scope of the technology is likely very broad, as MS detection is extremely versatile. The most direct applications are related to the high-throughput screening for other molecules that modulate the secretion of other peptide hormones such as Glucagon-like peptide 1 (GLP-1). In the context of the development of personalized medicine, our analysis process could be deployed to define the most appropriate treatment to restore the hormonal secretion of a patient suffering from T2D. This concept of searching for alternative therapeutics could naturally be extended to other disciplinary fields such as oncology or the study of neurodegenerative diseases. Nevertheless, the use of our process in a precise context remains subject to the prior knowledge of the protein(s) of interest.

Although encouraging, our study has several limitations. The use of the murine β-cell model Min6 is convenient, but translation to human cells remains to be undertaken. It may be relevant to confirm some of the target and/or small molecules identified in Min6 cells in the EndoC-BH1 human model, which has been shown to be useful for the identification of modulators of human beta-cell insulin secretion [13]. In addition, siRNA mediated knock-down may not be efficient enough to identify genes potentially involved in β-cell function. The use of CrispR/Cas9, as performed in human islets [14] or in EndoC-BH1 [15], may help to better identify genes that are directly involved in insulin secretion, strengthening the translation to human. Interestingly, a new generation of human beta cell line, EndoC-HB5 cells, is compatible with HTS experiments with a good reactivity to glucose stimulation.

## Figures and Tables

**Figure 1 cells-12-00849-f001:**
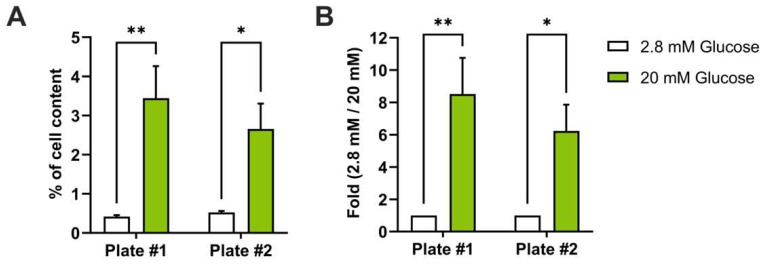
Automated GSIS is efficient to measure insulin secretion of Min6 cells. Min6 cells were cultured in 384-well plates and subjected to automated GSIS. (**A**,**B**) Insulin concentration was measured using ELISA and GSIS results were expressed as percentage of insulin content ((**A**), n = 5) or fold induction over 2.8 mM glucose ((**B**), n = 5). Statistical analyses were performed using two-way ANOVA with Bonferroni post-test analyses. Results are presented as means ± SEM. * *p* < 0.05; ** *p* < 0.01.

**Figure 2 cells-12-00849-f002:**
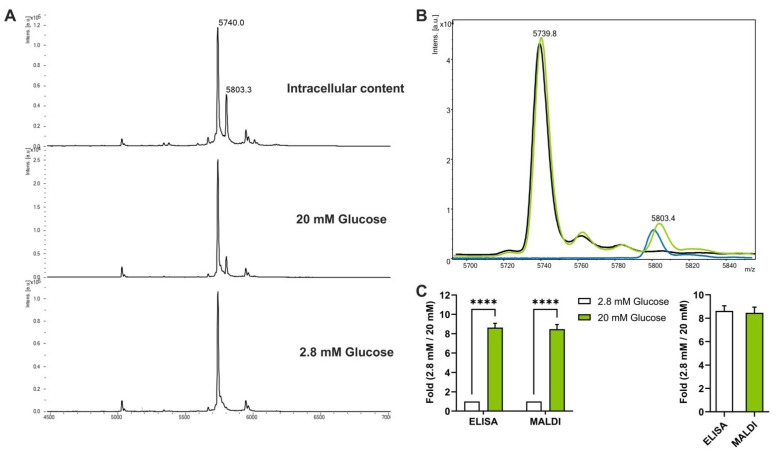
Measurement of insulin secretion from Min6 cells through MALDI-TOF mass spectrometry. (**A**) Mass spectra of the intracellular insulin content, 20 mM and 2.8 mM of glucose secretomes; (**B**) Superposition of the different mass spectra of murine and bovine insulin (black: bovine, blue: murine, green: mixture); (**C**) Quantification of insulin secretion after stimulation with glucose by MALDI-TOF mass spectrometry. Min6 cells were cultured in a 384-well plate and then subjected to automated GSIS and insulin secretion was measured comparing ELISA (n = 4) and MALDI-TOF (n = 16) as detection methods. Samples from two independent plates were analyzed. Statistical analyses were performed using two-way ANOVA with Bonferroni post-test analyses. Results are presented as means ± SEM. **** *p* < 0.0001.

**Figure 3 cells-12-00849-f003:**
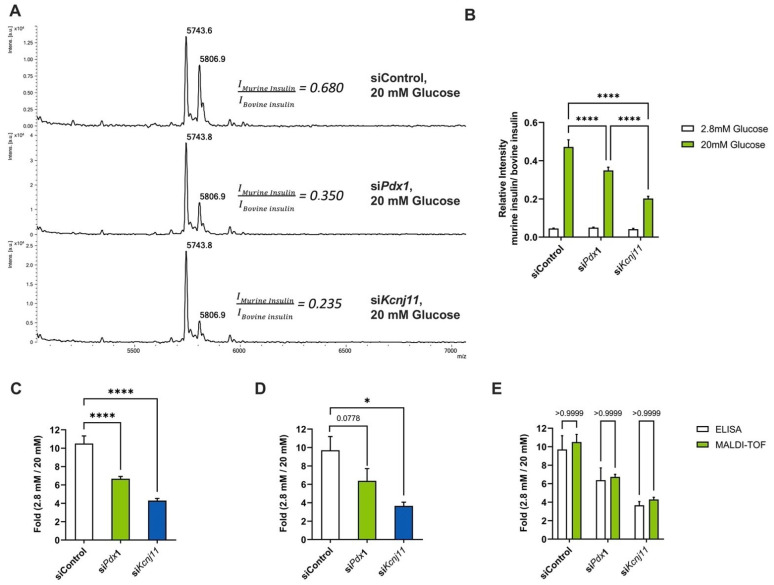
Automated GSIS of siRNA-transfected Min6 cells analyzed through MALDI-TOF mass spectrometry. (**A**) Mass spectra of the 20 mM secretomes of Min6 cells transfected with siControl, si*Pdx1* and si*Kcnj11*; (**B**) Quantification of mass spectra of Min6 cells transfected with siControl, si*Pdx1* and si*Kcnj11* subjected to GSIS, using relative intensities of the murine and bovine insulin spectrum area. (**C**,**D**) Glucose stimulation effect on insulin secretion presented as the fold of insulin secretion in 20 mM glucose over 2.8 mM Glucose. Insulin levels were measured from Min6 cells transfected by siControl, si*Pdx1* and si*Kcnj11* through MALDI-TOF mass spectrometry ((**C**), n = 16) or ELISA ((**D**), n = 4). (**E**) Comparison of quantification results obtained by mass spectrometry and ELISA assays. Statistical analyses were performed using two-way ANOVA with Bonferroni post-test analyses. Results are presented as means ± SEM. * *p* < 0.05; **** *p* < 0.0001.

**Figure 4 cells-12-00849-f004:**
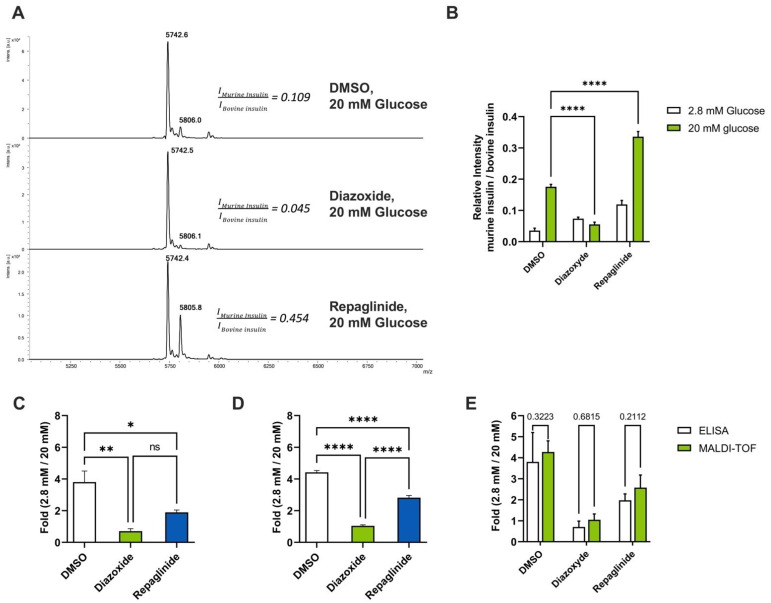
Min6 insulin secretion analysis of Min6 cells treated with diazoxide and repaglinide by MALDI-TOF mass spectrometry. (**A**) Mass spectra of the 20 mM secretomes of Min6 cells treated with DMSO, diazoxide 100 µM and repaglinide 100 nM; Quantification of insulin secretion after stimulation with glucose 2.8 mM and 20 mM of Min6 cells treated with DMSO, diazoxide 100 µM and repaglinide 100 nM (**B**,**D**) by MALDI-TOF mass spectrometry and (**C**) by ELISA. (**E**) Comparison of results obtained by mass spectrometry and ELISA assay. Two-way ANOVA with Bonferroni post-test analyses. Error bars are ± SEM and n = 4 for ELISA, n = 32 for MS. * *p* < 0.05; ** *p* < 0.01; **** *p* < 0.0001.

**Figure 5 cells-12-00849-f005:**
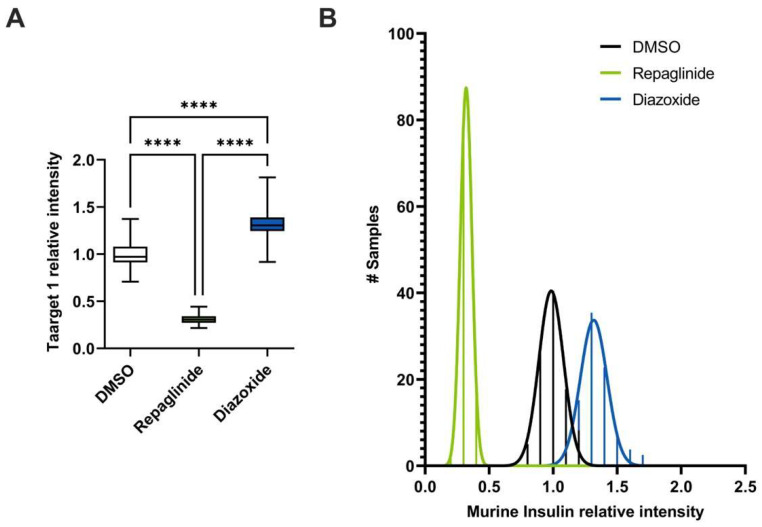
Analysis of insulin content of Min6 cells treated with diazoxide and repaglinide through MALDI-TOF mass spectrometry. (**A**) Quantification of intracellular insulin content of Min6 cells treated with DMSO, 100 µM diazoxide and 100 nM repaglinide through MALDI-TOF mass spectrometry. (**B**) Normal distribution of analysis of intracellular insulin content from Min6 cells treated with DMSO, repaglinide and diazoxide. Statistical analyses were performed using two-way ANOVA with Bonferroni post-test analyses. Results are presented as means ± SEM. n = 4 for ELISA, n = 160 for DMSO and n = 80 for repaglinide and diazoxide. **** *p* < 0.0001.

**Figure 6 cells-12-00849-f006:**
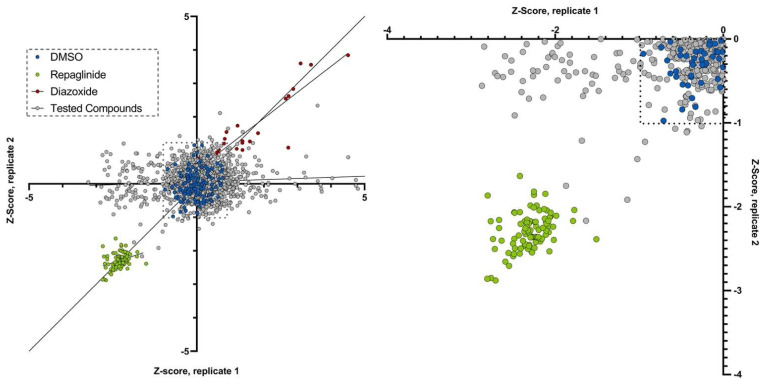
High-throughput chemical screen of Min6 cells identifies small molecules that may amplify insulin release and decrease insulin intracellular content after glucose stimulation. Min6 cells were treated with 1600 chemical compounds and insulin content was measured using MALDI-TOF mass spectrometry. Compounds from the chemical library are in grey. Repaglinide (green) was used as a positive control, whereas diazoxide (red) was used as an inhibitor of insulin secretion.

## Data Availability

Not applicable.

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
