# Peer review of "High-Throughput Quantitative Screening of Glucose-Stimulated Insulin Secretion and Insulin Content Using Automated MALDI-TOF Mass Spectrometry"

_cells, 2023, doi:10.3390/cells12060849_

Round 1

Reviewer 1 Report

Abstract

Line 28: The current worldwide number of people with diabetes is 537 million (Please see the 10th edition of IDF Atlas of Diabetes)

Introduction

Lines 46-47: Please rephrase “In Europe, the global prevalence of diabetes is currently estimated at 8 % of the population, with T2D representing 90 % of cases” to “rephrase “In Europe, the global prevalence of diabetes is currently estimated at 8 % of the population, with T2D representing about 90 % of cases”

Line 417: Please change “…Automated library screening to mesure…: to “….Automated library screening to measure..”

What is the probable advantage of Min6 cell line over others such as RIN, HIT, beta TC, MIN6 and INS-1etc.?

Reviewer 2 Report

In this study, the Authors describe an automated approach for the high-throughput screening of insulin secretion and content to identify new compounds or genes potentially involved in beta-cell function, using the murine beta-cell model Min6 and the MALDI-TOF mass spectrometry, in a 384-well plate format.

The topic of this manuscript is interesting to the field. However, I have some comments to clarify some data and results.

Major comments:

1.      Materials and Methods: KRB composition should be described (line 99); the described volumes are not consistent with those described in Fig. S2 and Results, lines 273-278 (lines 114-126); a reference on the use of endogenous cyclophilin mRNA levels to normalize qRT-PCR results in MIN-6 cells should be provided (line 238).

2.      Results: The Authors state “Therefore…the cells were washed 5 times with BSA-free KRB buffer supplemented with 2.8 mM glucose and incubated for 1 h at 37 °C. Then, half of the supernatant was collected… and the cell plates were complemented with KRB buffer containing glucose at 20 mM and incubated for 1 h at 37 °C. The high glucose fractions were then collected, and cells were lysed to measure insulin content” (lines 275-278). Was the remaining half of the supernatant removed before the cell plates were supplemented with KRB buffer containing 20 mM glucose? If not, what is the final glucose concentration achieved? Finally, was the entire volume of the high glucose fractions collected?

3.      Results: The Authors state “Min6 cells display a secretion rate which increases significantly after glucose stimulation, which represents 3 to 5 % of the total insulin content or a 6 to 8-fold increase of insulin secretion when Min6 cells were stimulated from 2.8 mM to 20 mM of glucose”. Are these results comparable to what has already been demonstrated in the literature on Min-6?

4.      Results (lines 304-307) “To quantitatively measure insulin from GSIS supernatants, samples were spiked with known concentrations of bovine insulin. Following MALDI-TOF mass spectrometry, a final mass spectrum was obtained… (Figure 2A)” and Legend of Fig. 2A (lines 323-324) “Mass spectra of the intracellular insulin content, 20 mM and 2.8 mM of glucose secretomes”: Do mass spectra refer to intracellular content of insulin or to insulin in the supernatants?  

5.      Legend of Fig. 2 (lines 327-328): Why is the number of experiments so different between ELISA (n=4) and MALDI-TOF (n=16)? After the initial high-throughput screening, do the Authors suggest validating the individual data obtained, for example by ELISA method?

6.      During an experiment with chemicals or siRNAs used for chronic times, if it results a reduction in the secreted or intracellular levels of insulin, how can it be excluded that these are not due to cell suffering/death? Wouldn't it be more appropriate to normalize the insulin values obtained with respect to the number of cells or the amount of total protein?

7.      Fig. S4: Do the Authors have data also for Kcnj11?

8.      The Authors should provide references to support the notion that increased insulin secretion leads to decreased protein content. For some compounds (used to stimulate pancreatic beta cells), in fact, this may not happen, because they could have simultaneous stimulation/inhibition effects on both insulin production and secretion, especially if they are used for chronic periods. This should be discussed. Consequently, the sentence “High-Throughput chemical screen of Min6 cells identifies small molecules that amplify insulin release and decrease insulin intracellular content after glucose stimulation” (lines 425-426) should be modified. Similarly, I believe that attention should be paid to the evaluation of insulin content as a sure indicator of insulin release in the results described in lines 430-443.

9.      What’s the differences between Fig. 5A and Fig. S5B?

Minor comments:

1. Legends of Supplementary Figures should be more informative.

3. Legend of Fig. 1 (lines 293-294): “* p<0.05; ** p<0.05” should be corrected.

5. In the graphs with “Fold” on the y axis, what the fold refers to should be indicated.

9. Fig. 4A: captions within graphics (“I murine insulin/I bovine insulin”) should be enlarged.

10. Fig. S6: “Expected data” on the y axes should be enlarged.

11. Fig. 5B should be described in the legend.

14. In suggesting the use of EndoC-BH1, their lack of incretin responsiveness should be mentioned (https://www.humancelldesign.com/human-beta-cell-lines/).

1. Typing errors: “fiazoxyle” (Materials and Methods, line 148); “MADI-TOF” (Results, lines 314, and Legend of Fig.6, line 427); “transfcetant” (Legend of Fig. S3); “after after” (Legend of Fig. S4); “trascriptomic” and “mesure” (Results, lines 344 and 417); “Glp-1)…)” (Discussion, line 477).

Reviewer 3 Report

n/a

Round 2

Reviewer 2 Report

The Authors satisfactorily answered many of my questions, clarifying the data and results, and making the appropriate changes to the manuscript.

A minor comment: data on the efficiency of siRNA targeting Kcnj11 could be included in the supplementary data or cited in the text as "unpublished data".